# Diversity among *Lasiodiplodia* Species Causing Dieback, Root Rot and Leaf Spot on Fruit Trees in Egypt, and a Description of *Lasiodiplodia newvalleyensis* sp. nov.

**DOI:** 10.3390/jof8111203

**Published:** 2022-11-15

**Authors:** Sherif Mohamed El-Ganainy, Ahmed Mahmoud Ismail, Zafar Iqbal, Eman Said Elshewy, Khalid A. Alhudaib, Mustafa I. Almaghasla, Donato Magistà

**Affiliations:** 1Department of Arid Land Agriculture, College of Agriculture and Food Sciences, King Faisal University, P.O. Box 420, Al-Ahsa 31982, Saudi Arabia; 2Plant Pests, and Diseases Unit, College of Agriculture and Food Sciences, King Faisal University, P.O. Box 420, Al-Ahsa 31982, Saudi Arabia; 3Vegetable Diseases Research Department, Plant Pathology Research Institute, Agricultural Research Center (ARC), Giza 12619, Egypt; 4Central Laboratories, King Faisal University, Riyadh 11451, Saudi Arabia; 5Department of Soil, Plant and Food Sciences, University of Bari A. Moro, 70126 Bari, Italy; 6Institute of Sciences of Food Production (ISPA), National Research Council (CNR), 70126 Bari, Italy

**Keywords:** elongation factor 1-alpha, β-tubulin, ITS, morphological characterization, pathogenicity phylogenetic analysis

## Abstract

*Lasiodiplodia* (family *Botryosphaeriaceae*) is a widely distributed fungal genus that causes a variety of diseases in tropical and subtropical regions. During 2020–2021, a routine survey of fruit tree plants was conducted in five Egyptian Governorates, and fresh samples exhibiting dieback, decline, leaf spot and root rot symptoms were collected. Collection from eight different symptomatic leaves, twigs, branches and roots of fruit trees yielded 18 *Lasiodiplodia*-like isolates. The sequencing data from the internal transcribed spacer region (ITS), partial translation elongation factor 1-alpha (*tef1-a*) and *β-tubulin* (*tub2*) were used to infer phylogenetic relationships with known *Lasiodiplodia* species. Two isolates obtained from black necrotic lesions on *Phoenix dactylifera* leaves were identified as a putative novel species, *L. newvalleyensis* sp. nov., and were thus subjected to further morphological characterization. The results of isolation and molecular characterization revealed that *L. theobromae* (*n* = 9) was the most common species on *Mangifera indica*, *Citrus reticulata*, *C. sinensis*, *Ficus carica*, *Prunus persica*, *Prunus armeniaca* and *Pyrus communis* trees. *Lasiodiplodia pseudotheobromae* (*n* = 5) was isolated from *M. indica*, *Prunus persica* and *C. sinensis*. *Lasiodiplodia laeliocattleyae* (*n* = 2) was isolated from *C. reticulata*. Pathogenicity test results suggested that all *Lasiodiplodia* species were pathogenic to their hosts. The present study is considered the first to characterize and decipher the diversity of *Lasiodiplodia* species associated with fruit trees in Egypt, using the multi-locus ITS, *tef1-a* and *tub2* sequence data, along with morphological and pathogenic trials. To our knowledge, this is the first report of *L. newvalleyensis* on *Phoenix dactylifera* and *L. laeliocattleya* on *C. reticulata* in Egypt and worldwide.

## 1. Introduction

The family *Botryosphaeriaceae* encompasses several fungal species that are found in all environmental and climatic zones of the world as endophytes or saprophytes pathogens [1]. *Lasiodiplodia* (family *Botryosphaeriaceae*) is a pluralistic genus distributed in tropical and subtropical areas that causes a variety of diseases, including cankers, dieback, fruit or root rot, branch blight, stem end rot and gummosis on a wide range of woody and fruit trees [1,2,3,4,5]. Since 2004 and until 2017, 43 species of *Lasiodiplodia* have been described [1,3,4,6]. Nonetheless, five new *Lasiodiplodia* species associated with blueberries have recently been discovered in China [7], bringing the genus *Lasiodiplodia* to forty-eight species. Members of the genus *Lasiodiplodia* exhibit diverse lifestyles on a wide range of host plants, ranging from endophytes, which cause asymptomatic infection on different plant tissues, pathogens, which cause diseases and saprophytes [1,8]. Among the *Lasiodiplodia* species, *L. theobromae* is a well-known plant pathogen associated with up to 500 hosts [9]. Diseases caused by species in the *Botryosphaeriaceae* have been reported since 1971 when *Botryodiplodia theobromae* was isolated from fruit rot and dieback of mango in Egypt. The fungal agent was later synonymized under *L. theobromae* and regarded as a causal pathogen for dieback on mango [3,10,11]; root rot on sugar beet dieback [12]; and canker and soft rot on other hosts, such as grapevine [13], walnut [14], maize [15], citrus [16], *Annona* spp. [17], *Phoenix dactylifera* [18], pome, stone fruit [19], *Citrus sinensis*, *C*. *aurantifolia* [20] and ornamental Ficus trees [21]. 

Characterization of *Lasiodiplodia* species has primarily relied on cultural and conidial characteristics and phylogenetic data [3,5,8,22,23,24,25]. Cultural and conidial characterization are often misleading and result in inaccurate identification due to overlapping in morphology [25,26]. Therefore, molecular characterization based on multi-locus sequence data has widely been applied to identify the *Lasiodiplodia* species, especially the *L. theobromae* species complex, which is difficult to distinguish based on morphology [1,8,23]. Recent multi-locus phylogenetic approaches using DNA sequence data of the internal transcribed spacers (ITS) of genomic rDNA [27], along with protein-coding genes such as translation elongation factor 1-alpha (*tef1-a*) and *β-tubulin* (*tub2*) [1,5,7,23], have aided in the identification of *Lasiodiplodia* species with strong phylogenetic support.

Based on the cosmopolitan presence of *Lasiodiplodia* species on various hosts and a very recent study [20], the distribution and prevalence of this fungal agent could be extended to other hosts in Egypt. In this sense, *Lasiodiplodia* species considered as a major pathogens occurring on a variety of hosts causing stem-end rot, fruit rot, decline, cankers and dieback. The current study was aimed at characterizing and deciphering the diversity of *Lasiodiplodia* species associated with wider fruit tree hosts in Egypt, using the ITS, *tef1-a* and *tub2* sequence data, together with morphological and pathogenic trials.

## 2. Materials and Methods

### 2.1. Sampling and Isolation

During 2020–2021, surveys of fruit tree plants, including *Mangifera indica*, *Citrus reticulata*, *Citrus sinensis*, *Ficus carica*, *Prunus persica*, *Prunus armeniaca, Pyrus communis* and *Phoenix dactylifera*, were conducted across five Egyptian Governorates: Beheira, Giza, Kaliobyia, Sharkia and New Valley (Appendix A). A total of fifty-seven symptomatic leaves, twigs, branches and roots of plants exhibiting leaf spot, dieback, decline and root rot symptoms were collected. Samples were subjected to pathogen isolation, as previously described [22]. The obtained *Lasiodiplodia*-like isolates and other associated fungi were cultured on potato dextrose agar (PDA) and stored at 5 °C in a refrigerator. The cultures were maintained in the culture collection facility at the Vegetable Diseases Research Department, Plant Pathology Research Institute, Agricultural Research Center (ARC), Giza, Egypt.

### 2.2. DNA Extraction and PCR Amplification

Genomic DNA was extracted from 5-day-old cultures of isolated fungi [28]. PCR amplification and sequencing of the ITS region of rDNA, including 5.8S, was performed using the primers ITS4 and ITS5 [27]. Part of the *tef1-α* region was amplified using EF1-728F and EF1-986R [29], and the tub2 region was amplified using Bt1a and Bt1b primers [30]. PCR amplifications were carried out in an ESCO Swift Maxi Thermal Cycler [31]. The resultant PCR amplicons were gel purified using the CloneJet PCR cloning kit (ThermoFisher Scientific, Waltham, MA, USA) and sequenced in both directions using Sanger sequencing at Macrogen Inc. (Seoul, Korea). Sequences obtained in this study were deposited in GenBank database, and their accession numbers were obtained (Table 1).

### 2.3. Phylogenetic Analyses

MEGA XI (version 11.0.8) was used to trim and edit the obtained ITS, *tef1-α* and *tub2* sequences to remove ambiguous ends from both directions [32]. MAFFT version 7 was used to assemble and align the sequences with the closely related *Lasiodiplodia* spp. [33]. Sequences were retrieved from the NCBI GenBank database (http://www.ncbi.nlm.nih.gov, accessed on 25 July 2022). Phylogenetic analysis was conducted using PAUP version 4.0a [34]. Maximum parsimony (MP) analysis was conducted using the heuristic search option with random stepwise addition based on 1000 replicates, tree bisection and reconnection (TBR) as branch swapping algorithms, and random taxon addition sequences for the construction of MP trees. Branches of zero length were collapsed, and all multiple equally parsimonious trees were saved. MAXTREES was set to 10,000. In the analysis, all characters were unordered and had equal weight; gaps were treated as missing data. Tree length (TL), consistency index (CI), rescaled consistency index (RC), retention index (RI) and the homoplasy index (HI) were calculated for parsimony [35]. The phylogenetic relationship was inferred with 1000 bootstrap replicates and included 104 sequences, representing 103 of *Lasiodiplodia* species, and a *Diplodia mutila* (CMW 7060) sequence as an outgroup taxon (Table 1). Bayesian analysis was performed using MrBayes v3.2.7a [36] on Cipres Science Gateway (www.phylo.org, accessed on 25 July 2022) [37], on the combined, partitioned dataset with the substitution models, calculated for each partition, by ModelFinder on IQ-TREE multicore version 2.2.0 [38,39]. Bayesian analysis was run in duplicate with four Markov chain Monte Carlo (MCMC) chains, with random trees for 10,000,000 generations, sampled every 1000 generations. The temperature value was lowered to 0.10, burn-in was set to 0.25 and the run was automatically stopped when the average standard deviation of split frequencies ended up below 0.01. A total of 4222 trees were read in the two runs, 2111 each, and 25% of trees were discarded in each run as the burn-in phase of the analysis. Posterior probabilities were determined from a consensus tree generated from the remaining 1584 trees of each run. Maximum likelihood (ML) analysis was computed with IQ-TREE multicore version 2.2.0, setting ModelFinder + tree reconstruction + ultrafast bootstrap based on 10,000 replicates [39,40,41]. The phylogenetic trees of the MP, ML and BP were viewed in FigTree version 1.4.4 (http://tree.bio.ed.ac.uk/software/figtree, accessed on 25 July 2022).

### 2.4. Morphological Examination

Fungal structures were examined by inducing sporulation on 2% water agar (WA) medium supplemented with double-autoclaved pine needles, as described by Ismail et al. [3]. A 5-mm mycelial plug from each isolate was placed in the center of WA plates and incubated for 10–20 days at 25 ± 2 °C near direct light with a 12 h photoperiod. Sections were made through conidiomata using Leica CM1100 microtome and mounted in lactic acid. Measurements were done for 30 conidiogenous cells, 30 paraphyses and 50 conidia from material mounted in water. Fungal structures were imaged with a Nikon Coolpix 995 digital camera connected to a Leica, DM 25,000 LED microscope. Colony morphology was observed on PDA medium after 7 days of incubation at 25 °C in the dark. 

### 2.5. Evaluation of Temperature’s Effect on the Mycelial Growth

The effects of different temperature on the mycelial growth of *L. newvalleyensis* were investigated. Three plates for each temperature were inoculated with 6-mm plugs from the actively margins of 5-days-old cultures in the center of the 85-mm PDA. Petri dishes were incubated in the dark at 6 different temperatures (10, 15, 20, 25, 30 and 35 °C). After 3 days, colony diameters were determined, and the data were converted to radial growth in millimeters. 

### 2.6. Pathogenicity Test on Seedlings and Leaves

*Lasiodiplodia* isolates were tested for their pathogenicity against their hosts of origin. Pathogenicity was determined in 6–10-month-old seedlings of *Citrus reticulata*, *M. indica*, *Prunus persica*, *Prunus armeniaca* and *Pyrus communis*. Apparently healthy leaves of *Citrus reticulata*, *F. carica*, *M. indica* and *Phoenix dactylifera* were selected for pathogenicity. Three replicates were used, and each replicate consisted of three leaves, meaning a total of 12 leaves were used for each isolate. *Lasiodiplodia* isolates were plated on PDA for 5-days at 25 ± 2 °C in the dark prior to inoculation [3,22]. Inoculations of seedlings and leaves were performed according to Ismail et al. [3,22]. Three replicates were used per isolate, and each replicate comprised three plants with a total of 12 seedlings for each isolate. The inoculated plants were maintained under greenhouse conditions at 25 ± 2 °C and 70–80% relative humidity, and examined periodically for symptom development. The trials were arranged in a completely randomized factorial design, and the trials were repeated once. After 30 days, the pathogenicity of the tested isolates was terminated, and the results were recorded as the extent of necrotic lesions (in centimeters) developed around the inoculation sites for seedlings and leaves. The dimensions of the inoculated wounds were not subtracted from final measurements. Values were transformed by Log^2^ for analysis and separation of means. Re-isolation of the tested isolates was performed from the margins of the necrotic lesions on PDA medium amended with streptomycin sulfate (0.1g L^−1^) and incubated in the dark at 25 ± 2 °C.

### 2.7. Data Analysis 

The obtained data were subjected to one-way ANOVA [42]. The data of lesion lengths were not normally distributed and were then log transformed. Mean values of the transformed lesion diameters (cm) and mycelial growth (mm) were compared using the least-significant difference (LSD) test at (*p <* 0.05). The statistical program SPSS 8.0 was used to analyse the data.

## 3. Results

### 3.1. Symptoms, Isolation and Frequency 

Several symptom patterns on different fruit tree organs were observed, but the most prevalent disease phenotype was dieback and decline. On mango trees, stem cracking symptoms with black liquid oozing from infected tissues were also observed (Figure 1A). Other symptoms observed included dieback of the young twigs starting from the tip and extending downward (Figure 1B), infected twigs (cross section) showing brown vascular discoloration of tissues on one side (Figure 1C), brown to black lesions on the leaf margins (Figure 1D), root rot of mango seedlings (Figure 1E), black lesions under cambium tissues of the crown area (Figure 1F) and apical part of roots (cross section) showing brown discoloration of internal tissues (Figure 1G).

On *Prunus persica* trees, the observed symptoms were dieback of the young twigs and branches starting from the tip and extending downward (Figure 2A); infected twigs (cross and longitudinal sections) showing the brown vascular discoloration of tissues in one side (Figure 2B–D); brown and root rot, especially on old trees (Figure 2E); and brown discoloration under cambium tissues of the crown area (Figure 2F). The symptom on *Pyrus communis* trees was dieback of the young twigs starting from the tip and extending downward (Figure 3A). Cross-sections of infected twigs to compare the infected and healthy tissues also showed brown vascular discoloration of tissues on one side (Figure 3B,C). It was possible to observe dieback and decline of the young twigs and branches of *Prunus armeniaca* starting from the tip extending downward (Figure 3D) and dieback of the branches on one side, giving V-shape symptoms (Figure 3E,F). Lesions with different appearances were observed on *C. reticulata*: large necrotic black lesions starting from the leaf margins and inside the leaf blades (Figure 4A,B). In addition, dieback symptoms were observed on young twigs of *C. reticulata* (Figure 4C) and *C. sinensis* (Figure 4D). Furthermore, brown to black lesions were recorded on the young leaves of *F. carica* (Figure 4E,F) as well as on the leaves of *Phoenix dactylifera* (Figure 4G,H). A total of 18 *Lasiodiplodia*-like isolates (growing fast on medium, with a greenish brown to dark greyish blue mycelium) and other associated fungi (4 isolates of *Alternaria* spp., 2 isolates of *Cladosporium* spp. and 2 isolates of *Pestalotiopsis* spp.) were isolated from eight different fruit trees from five Egyptian Governorates. In total, 18 *Lasiodiplodia*-like isolates were isolated—4 from branches, 7 from leaves, 4 from twigs, 2 from roots and 1 from stem cracking (Appendix A). All isolates were included in the phylogenetic study.

### 3.2. Phylogenetic Analyses

The sequences of the three gene regions were combined, yielding a dataset consisting of 1114 characters (ITS: 482 bps; tef-1α: 274 bps; tub2: 358 bps), including gaps of 104 *Lasiodiplodia* taxa (Appendix A). Of these characters, 72 characters were parsimony-uninformative, 156 were parsimony-informative and 886 (proportion = 0.795) were constant. Heuristic search with the random addition of taxa (1000 replicates) resulted in the phylogenetic tree (TL = 445 steps, CI = 0.633, RI = 0.868, RC = 0.550, HI = 0.366) and the most parsimonious tree is presented in Figure 5. The topology of the tree generated by MP analysis was congruent with the 50% majority-rule consensus tree. The phylogenetic tree generated by ML analysis based on the combined ITS, *tef-1α* and *tub2* sequence alignments is presented in Figure 6. Based on the ITS, *tef-1α* and *tub2* dataset, ML analysis revealed that *Lasiodiplodia* isolates can be grouped into five major clades. Among all, five isolates belong to clade containing *L. pseudotheobromae* (CBS116459 and CGMCC3 18047), as highly supported by the bootstrap (BS)/posterior probability (PP) values of 98/0.92%. Most of the isolates (nine isolates) grouped with *L. theobromae* (CBS111530 and CBS164.96) in a clade, which was strongly supported with BS/PP values of 84/0.91% (Figure 6). Additionally, two isolates clustered with *L. laeliocattleyae* (CBS130992) in a clade, which was supported with strong values of BS/PP, 100/1.0%. Notably, two isolates, EGY20113 and EGY20114, of *L. newvalleyensis*, representing a potential novel species grouped together in an distant clade, which was supported with BS/PP 93/0.91%, sister to a clade containing *L. exigua* BL104 and *L. americana* CERC1961, that highly supported with BS/PP 100/1.0% and to a clade containing *L. mahajangana* CMW27801 and CMW27818, which was supported with BS/PP 99/1.0%. 

### 3.3. Taxonomy

***Lasiodiplodia newvalleyensis*** A.M. Ismail, S.M. El-Ganainy and E.S Elshewy, sp. nov (Figure 6).

MycoBank: MB843771.

The etymology refers to the place New Valley Governorate from where this species was isolated.

Sexual morph: Absent. Asexual morph; *Conidiomata* (Figure 7b) produced on pine needles on WA within 10–15 days; mostly solitary or in aggregates; dark-grey to black; globose to subglobose; covered with dense hairy mycelium; semi-immersed; becomes erumpent when mature. A vertical section through pycnidia shows outer layers of pycnidia composed of approximately 4–8 dark-brown, thick-walled cells layers of *textura angularis*, followed by hyaline thin-walled cells towards the centre (Figure 7c). *Paraphyses* (Figure 7d,e), hyaline and subcylindrical, arise between the conidiogenous cells. They are aseptate, wider at the base, slightly swollen at the apex, 14.9–44.5 µm long and 1.9–3.7 µm wide. *Conidiophores* reduced to conidiogenous cells. *Conidiogenous cells* (Figure 7f,g) are holoblastic, thin-walled hyaline, cylindrical and sometimes swollen slightly at the base. They have a rounded apex, proliferate recurrently to produce 1–2-minute annelations, are 4.6–10.5 µm long and are 3.2–5 µm wide. *Conidia* (Figure 7h–k) are initially hyaline, smooth, thick-walled, aseptate and obovoid to ellipsoid, contain granular contents and are mostly round at both ends; they have the same form when mature. Conidia become brown, are septate with 1-septum, have longitudinal striations and measure 17.2–26.7 × 10.5–13.3 µm (av. of 50 conidia ± SD = 22 ± 1.8 µm long, 11.7 ± 0.7 µm wide, L/W ratio = 1.8).

*Cultural characteristics* (Figure 7a): Colonies raised on a mycelium mat were moderately dense, and initially white to smoke-grey but turned greenish grey on the front side and greenish grey on the reverse side. The colour becomes dark slate blue with age. Pycnidia was produced on PDA after 7 days under the above-mentioned conditions. Colonies reached the edge of the Petri plate, 85 mm, after 3-days in the dark at 30 °C. Cardinal temperature requirements for growth: minimum, 15 °C; maximum, 35 °C; and optimum, 30 °C (Figure 8). No growth was observed at 10 °C. Isolates produced a pink pigment in PDA medium at 35 °C.

*Materials examined*: Egypt, New Valley Governorate—large dark-brown lesions on leaves of date palm trees (*Phoenix dactylifera*), May 2020, A.M. Ismail, (holotype; a dry culture on pine needles: EGY H-240483); living culture ex-type: EGY20114.

**Notes:***Lasiodiplodia newvalleyensis* is phylogenetically distinct from other species of *Lsiodiplodia*. It forms a basal clade comprised of *L. nanpingensis*, *L. mahajangana*, *L. curvata*, *L. irregularis*, *L. pandanicola*, *L. magnoliae*, *L. chonburiensis*, *L. caatinguensis*, *L. exigua* and *L. americana*. Morphologically, the unbranched and shorter paraphyses (14.9– 44.5 × 1.9–3.7 µm) of *L. newvalleyensis* make the latter distinct from *L. nanpingensis* (102 × 3.5 µm) [7], *L. caatinguensis* (31.1–60.2 × 2.1–5.0 μm) [5] and *L. exigua* (66 × 5 µm) [43]. Furthermore, the aseptate paraphyses of *L. newvalleyensis* distinguished it from 1-septate *L. irregularis* [44] and from *L. mahajangana* [45]. The curved shape of conidia of *L. curvata* distinguished it from *L. newvalleyensis* [44]. Moreover, *L. newvalleyensis* have longer conidia (17.2–26.7 × 10.5–13.3 µm) than *L. caatinguensis* (13–20.2 × 10.1–12.5 μm) [5]. In addition, the conidia dimensions of *L. newvalleyensis* (17.2–26.7 × 10.5–13.3 µm) are distinguishable from those of *L. pandanicola* (14–38 × 9–22 µm) [46] and *L. magnoliae* (24–30 × 11–15 μm) [47]. The conidia shape (obovoid to ellipsoid) and dimensions (17.2–26.7 × 10.5–13.3 µm) of *L. newvalleyensis* are also distinguishable from those of *L. chonburiensis* that has subglobose to oval conidia with dimensions 23 × 12 µm [46]. *Lasiodiplodia newvalleyensis* and *L. americana* share almost the same conidia characteristics; however, the later differs by its longer (90 × 2–3.5 µm) and 1–3-septate paraphyses [48].

### 3.4. Pathogenicity Tests on Seedlings and Leaves

Pathogenicity tests revealed that all isolates were pathogenic to their hosts of origin to different degrees of severity. The control plants exhibited small zones of necrotic tissues due to wound reaction. Not all *Lasiodiplodia* isolates from the same species reacted in the same manner on the tested hosts. There was significant (*p* < 0.05) variation between isolates of *L. theobromae* and *L. pseudotheobromae* in terms of lesion length (Figure 9A). Out of all *L. theobromae* isolates, only EGY2082 and EGY2042 were aggressive on *Mangifera indica*, producing the largest lesions measuring 6.33 and 5.65cm (Figure 9A). EGY2048 was the most aggressive among *L. pseudotheobromae* isolates, causing lesions of 6.26 cm on *Prunus persica* (Figure 9A). The remaining *L. theobromae* and *L. pseudotheobromae* isolates induced smaller lesions that were not significantly different according to the LSD test (*p* < 0.05). Some isolates (EGY2048, EGY2082 and EGY20100) induced typical dieback symptoms on *Mangifera indica* in the early stage of infection, which progressed further with the fungal growth (upward and downward) and led to wilting and drying of the apical part and the terminal leaves, giving the scorched appearance (Figure 10A). The *L. theobromae* isolate (EGY2082) was pathogenic to *F. carica* and induced necrotic tissues similar to those observed on the origin host (Figure 10B). Both *L. laeliocattleyae* isolates (EGY2033 and EGY2038) were pathogenic to *C. reticulata* leaves (Figure 10B) with average lesion lengths of 3.27 and 3.49 cm, respectively, and were not statistically different (*p <* 0.05) from each other (Figure 9B). Additionally, the two isolates (EGY20113 and EGY20114) of the novel *L. newvalleyensis* species were highly pathogenic to *Phoenix dactylifera* leaves (Figure 10D,E) and produced lesions with average diameters of 4.44 and 3.91 cm, respectively (Figure 9B). 

## 4. Discussion

Based on the results of the current study, four species of *Lasiodiplodia* associated with diseases on different fruit trees were isolated and characterized. These were identified as *L. theobromae*, *L. pseudotheobromae*, *L. laeliocattleya* and the newly recognized species *L. newvalleyensis*. The new species was distinguished from other taxa in *Lasiodiplodia* based on the phylogenetic inferences of the ITS, *tef1-α* and *tub2* and morphological characteristics. To our knowledge, this is the first report of *L. newvalleyensis* causing leaf lesions on *Phoenix dactylifera* in Egypt and worldwide.

*Lasiodiplodia* species do not only occur as latent endophytes in asymptomatic plants, but are also associated with different symptoms occurring on a variety of hosts, including stem-end rot, fruit rot, decline, cankers and dieback [3,49]. In Egypt, *L*. *theobromae*, previously known as *Botryodiplodia theobromae*, was considered as the main causal agent of fruit rot and dieback of mango [10]. In the current work, *L. theobromae* was the most commonly isolated species causing different kinds of symptoms on *M. indica*, *C. reticulata*, *C. sinensis*, *F. carica*, *Prunus persica* and *Pyrus communis* trees. This finding is supported by previous studies which showed that *L. theobromae* has the ability to target a wide variety of fruit and woody trees plants in Egypt [18,19], along with ornamental *Ficus* trees [21]. *Lasiodiplodia theobromae* was also reported to cause gummosis and dieback of *Prunus persica* in Egypt [50]. Very recently, *L. theobromae* was reported as a causal agent of dieback, branch cankers and gummosis on *C. sinensis* and *C. aurantifolia* in Egypt [20]. Similar results were reported, and *L. theobromae* was the most frequently isolated from *M. indica* in Western Australia and Brazil [51,52].

In our study, *L. pseudotheobromae* ranked second in terms of isolation frequency and was associated with leaf lesions and dieback of *M. indica* and *C. sinensis*, along with root rot on *Prunus persica*. This species has a worldwide distribution and causes mainly stem-end rot, dieback and cankers on a wide range of hosts [3,4,5,24,25,49,53,54,55,56]. It was reported to cause dieback in only mango trees in Egypt [3]. However, the current study reported the presence of *L. pseudotheobromae* on other trees in Egypt. Reports on various hosts in different geographical areas suggested that *L. pseudotheobromae* has a wide host range and that its distribution might extend to other plant hosts and areas [45]. The low frequently with which *L. laeliocattleya* was isolated from *C. reticulata* suggests that this species has a limited geographical distribution. However, it has previously been reported to be on mango trees in Egypt [3] and Peru [57] and on coconut and mango trees in Brazil [52,58]. 

The extensive phylogenetic Inference based on multiple gene sequences has played an important role in delimiting novel species in the genus *Lasiodiplodia* [7,25,59]. In this study, the use of combined ITS, *tef1-α* and *tub2* sequence data enabled us to resolute the single cryptic species within *L. theobromae* species complex and provide novel clues into taxonomic novelties. The newly identified species was named as *L. newvalleyensis*, and its morphological description is supplemented. Several studies have demonstrated that using a single gene region is insufficient to delimit cryptic species [60,61,62], and therefore, to resolve species boundaries in the genus *Lasiodiplodia*, more than one gene region is required. This approach has revealed the presence of cryptic species in several genera in the family *Botryosphaeriaceae*. The multi-locus sequence data of ITS, *tef1-α* and *tub2* were used to separate *Lasiodiplodia* species in this study. Several studies have relied on morphological characteristics such as conidia dimensions, morphology and morphology; the sizes of paraphyses; and DNA sequence data for identifying *Lasiodiplodia* species [7,44,46,47,48]. However, several morphological features can overlap [25,26,63] but are still complimentary tools when combined with DNA phylogeny to distinguish new species in *Botryosphaeriaceae*. In this study, the shapes and lengths of paraphyses were used to differentiate *L. newvalleyensis* from the closely related species (Figure 7). Burgess et al. relied on the septation of paraphyses to discriminate between *Lasiodiplodia* spp. and indicated that *L. crassispora*, *L. gonubiensis* and *L. venezuelensis* have septate paraphyses, whereas other species are aseptate [64]. However, in this study, septate paraphyses were observed for *L. pseudotheobromae*, as previously reported by Alves et al. [56]. Using a similar approach, Damm et al. distinguished *L. plurivora* from *L. crassispora* and *L. venezuelensis* based on the morphology of the paraphyses [65]. This was also followed by a study of Abdollahzadeh et al. who distinguished *L. gilanensis* from *L. plurivora* and *L. hormozganensis* from *L. parva* and *L. citricola* using the morphology of the paraphyses [25]. In addition, Ismail et al. relied on the morphology of the paraphyses to distinguish *L. laeliocattleya* from the phylogenetically related *L. hormozganensis* [3]. 

Culture characteristics have also played a role in distinguishing *Lasiodiplodia* species. Alves et al. discriminated *L*. *parva* and *L*. *pseudotheobromae* from *L*. *theobromae* based on the production of a pink pigment in culture [56]. In contrast, the findings of Abdollahzadeh et al. revealed that *L*. *theobromae* and other *Lasiodiplodia* species, with the exception of *L*. *hormozganensis,* produce pink pigment on PDA at 35 °C [25]. In the present study, *L. newvalleyensis* produced a dark-pink pigment in PDA after 4 days at 35 °C; the color become darker with age. Colonies of *L. newvalleyensis* covered the 90 mm plates after 3 d at the optimum temperature of 30 °C. This finding is supported by those reported in previous studies that the optimum growth temperature for *Lasiodiplodia* species ranges between 25 and 30 °C [66,67]. Moreover, *L. newvalleyensis* could not grow at 10 °C, which is in contrast with the observations made by Alves et al. [56] and those of Abdollahzadeh et al., who found that all studied *Lasiodiplodia* isolates grow at the same temperature [25]. Our results are corroborated by those of a study on the mycelial growth of *L. viticola*, which could not grow at 10 °C [68]. However, the recently described novel species *L. guilinensis*, *L. huangyanensis*, *L. linhaiensis* and *L. ponkanicola* showed the ability to grow at 10 °C [67]. Thus, culture characteristics are of limited value in species determination due to their variation between isolates of a given species.

All *Lasiodiplodia* species showed the ability to spread through the internal tissues above and below the points of inoculation, causing brown to black necrotic lesions (Figure 10). The upward and downward progression inside the apparently healthy tissues reflected the well-known endophytic nature of these fungi [68,69,70,71]. In our study, we could not compare the severity of certain species on their hosts due to the low number of isolates recovered from the same hosts. This was evident for the single isolates of *L. theobromae* obtained from *Pyrus communis*, *M. indica*, *Prunus armeniaca*, *C. reticulata* and *F. carica*. There was significant (*p* < 0.05) variation within isolates of *L. pesudotheobromae* and *L. theobromae* in terms of severity. Variation in severity among *L. theobromae* and *L*. *brasiliensis* was also reported [72]. Recent findings confirmed that isolates of *L. theobromae* are more virulent than *D. seriata* on grapevines in Mexico [73]. Our results indicated that *L. theobromae* was more aggressive than *L. pesudotheobromae*, which induced the largest lesions and severe dieback symptoms on *M. indica.* These results are in contrast with those obtained by Ismail et al., who demonstrated that *L. pesudotheobromae* was highly pathogenic to *M. indica* than *L. theobromae* [3]. Furthermore, Leala et al. confirmed that *L. pesudotheobromae* and *L. theobromae* are pathogenic to acid lime and valencia orange [20]. Therefore, the high-frequency isolation, together with the results of pathogenicity, led us to consider that *L. pesudotheobromae* and *L. theobromae* are important fungal pathogens in Egypt. The low incidence, together with the fact that the only two isolates of *L. laeliocattleya* induced the smallest lesions on *C. reticulata*, suggest that this species is of a little importance and does not contribute significantly to citrus diseases. Our implications are based on earlier reports which demonstrated that *L. mahajagana* was not a primary pathogen due to its low incidence and virulence on *Terminalia catappa* [45], and *Fusicoccum bacilliforme* is a weak pathogen on mango plants due to its low isolation frequency and the small lesions it produces on mango plants [74]. A recent study also confirmed our suggestion that only *L. pesudotheobromae* and *L. theobromae* have been reported on citrus in Egypt [20]. Likewise, it was stated that species of *Lasiodiplodia* were more virulent against citrus, *L. pesudotheobromae* being the most widely distributed in China [73]. The two isolates of the newly described species *L. newvalleyensis* showed pathogenic ability on the leaves of *Phoenix dactylifera*, and there was no significant (*p* < 0.05) difference among them in terms of severity [66]. 

To conclude, the studies demonstrated here added a new species and two new host records to the list of *Lasiodiplodia* species. Therefore, this is the first report of *L. laeliocattleya* on *C. reticulata* and *L. newvalleyensis* on *Phoenix dactylifera* in Egypt and worldwide. The *L. laeliocattleya* and the newly described species *L. newvalleyensis* might pose a major threat to citrus and date palm cultivations and other fruit trees in the reported area. Therefore, further studies are needed, including extensive surveys and pathogenicity assays to clarify the ecology and to highlight their relative roles in causing diseases on other hosts. The external and internal symptoms developed by *Lasiodiplodia* species can evidently reflect the capacity of inoculated fungi to cause diseases and to spread rapidly throughout the vascular tissues, even if their hosts are not subjected to stress factors.

## Figures and Tables

**Figure 1 jof-08-01203-f001:**
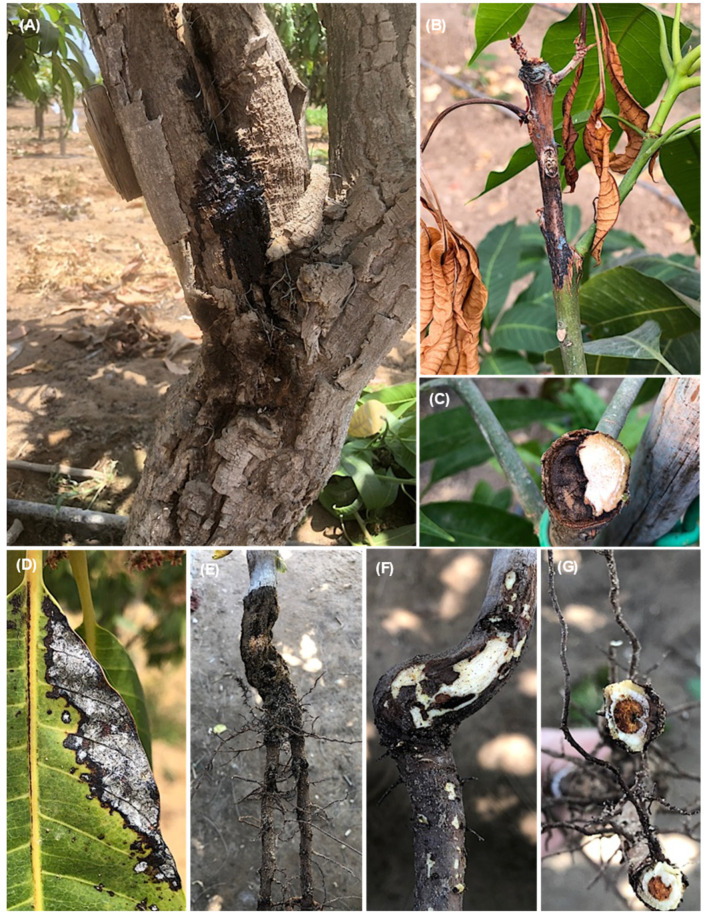
Symptoms observed on *M. indica* plants included stem cracking and gummosis (A) and dieback of the young twigs starting from the tip and extending downward (**B**). Cross-section of infected twigs showing the brown vascular discoloration of tissues in one side (**C**). Brown to black lesions on the leaf margins of the affected leaves (**D**). Root rot of mango seedlings (**E**). Black lesions under cambium tissues of the crown area (F), and cross-section of an apical part of roots showing brown discoloration of internal tissues (**G**).

**Figure 2 jof-08-01203-f002:**
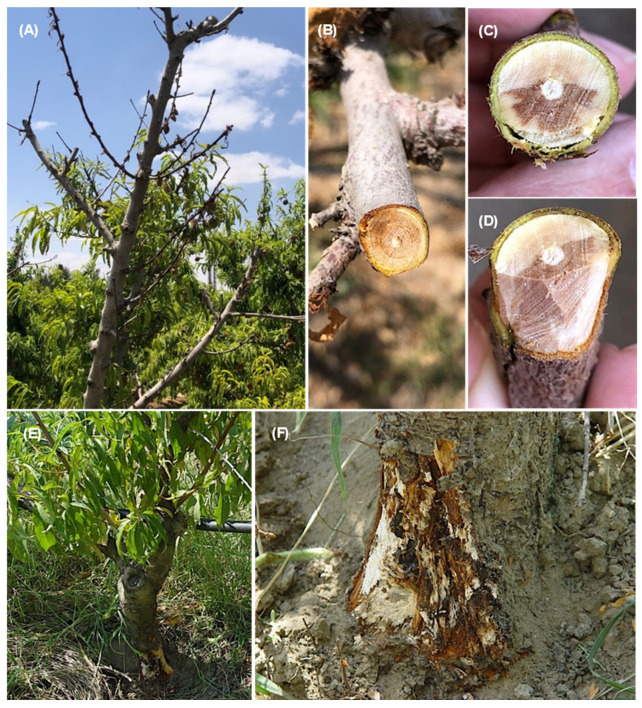
Symptoms observed on *Prunus persica* plants included dieback of the young twigs and branches starting from the tip and extending downward (**A**); cross (**B**,**C**) and longitudinal (**D**) sections of infected twigs showing the brown vascular discoloration of tissues in one side, crown and root rot (**E**); brown discoloration under cambium tissues of the crown area (**F**).

**Figure 3 jof-08-01203-f003:**
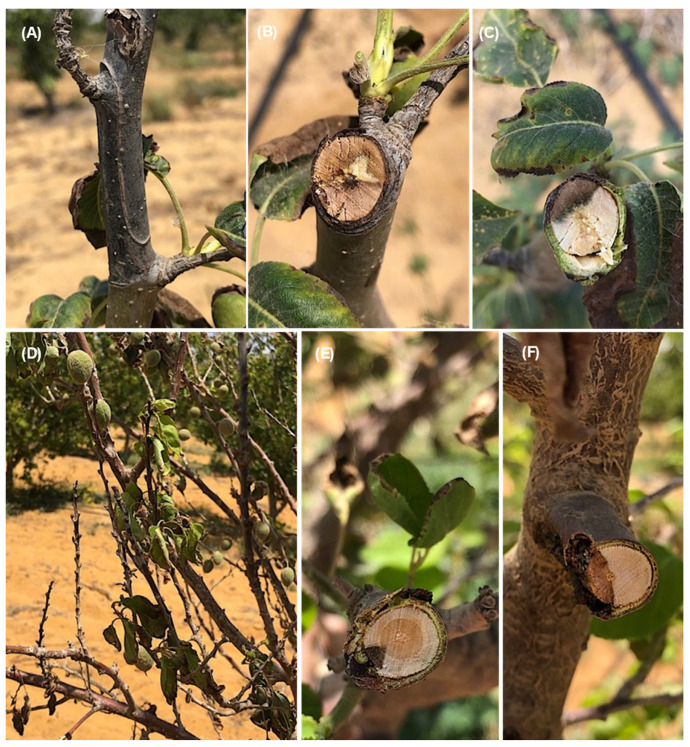
Symptoms on *Pyrus communis* plants included dieback of the young twigs starting from the tip and extending downward (**A**); cross-sections of infected twigs showing the brown vascular discoloration of tissues on one side (**B**,**C**); dieback and decline of the young twigs and branches of *Prunus armeniaca* starting from the tip and extending downward (**D**); dieback of the branches on one side, giving V-shape symptoms (**E**,**F**).

**Figure 4 jof-08-01203-f004:**
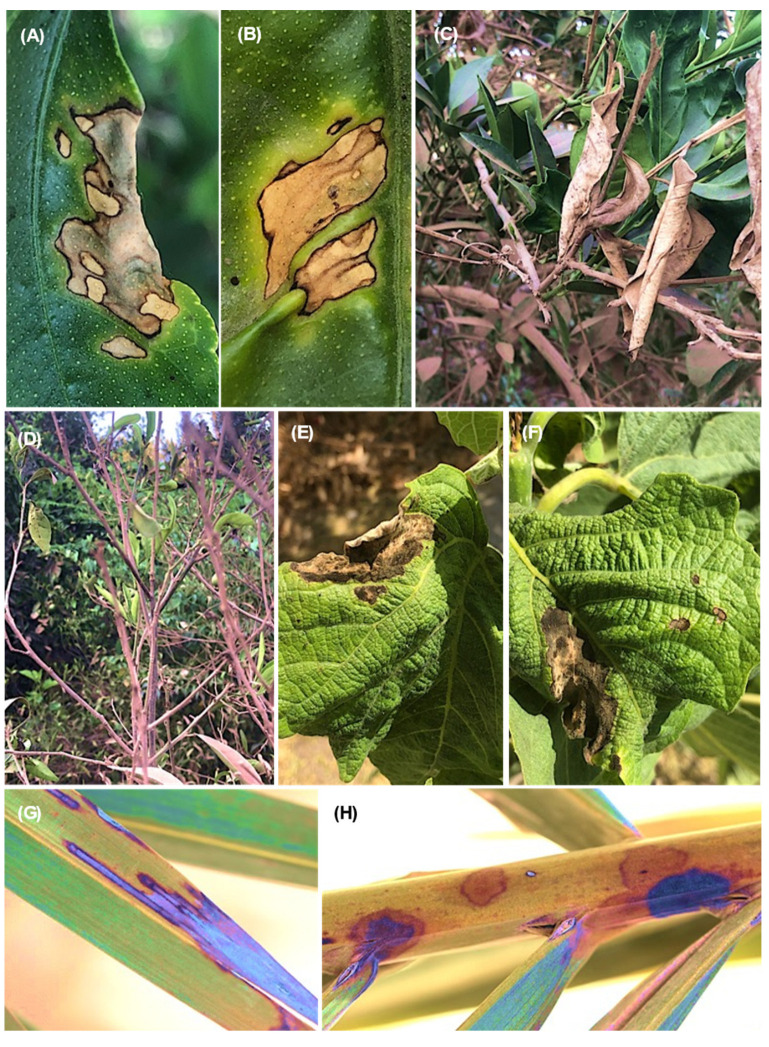
Symptoms on *C. reticulata*: large necrotic black lesions that start from the leaf margins and inside leaf blade (**A**,**B**); dieback on young twigs of *C. reticulata* (**C**) and *C. sinensis* (**D**); brown to black lesions on *F. carica* (**E**,**F**) and on *Phoenix dactylifera* (**G**,**H**).

**Figure 5 jof-08-01203-f005:**
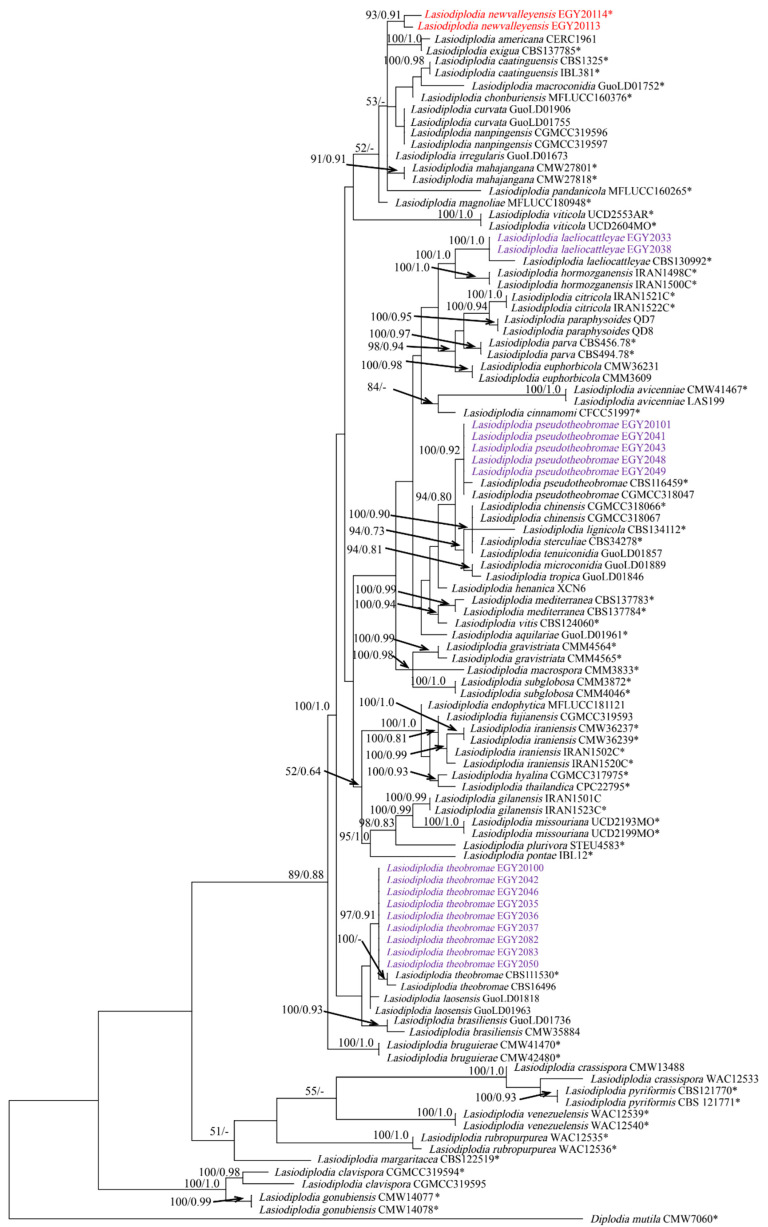
Phylogenetic tree based on maximum parsimony analysis (MP) through heuristic searches of the combined ITS, *tef-1α* and *tub2* dataset of *Lasiodiplodia* species. Branches are shown on nodes with bootstrap values (BS %) and Bayesian posterior probabilities (PP). Branches not supported with BS or PP are marked with –, and isolates representing ex-type are marked with *. *Diplodia mutila* CMW 7060 was used as an outgroup taxon to validate the tree. The isolates obtained in this study are blue, and those newly described and ex-type species are in red boldface.

**Figure 6 jof-08-01203-f006:**
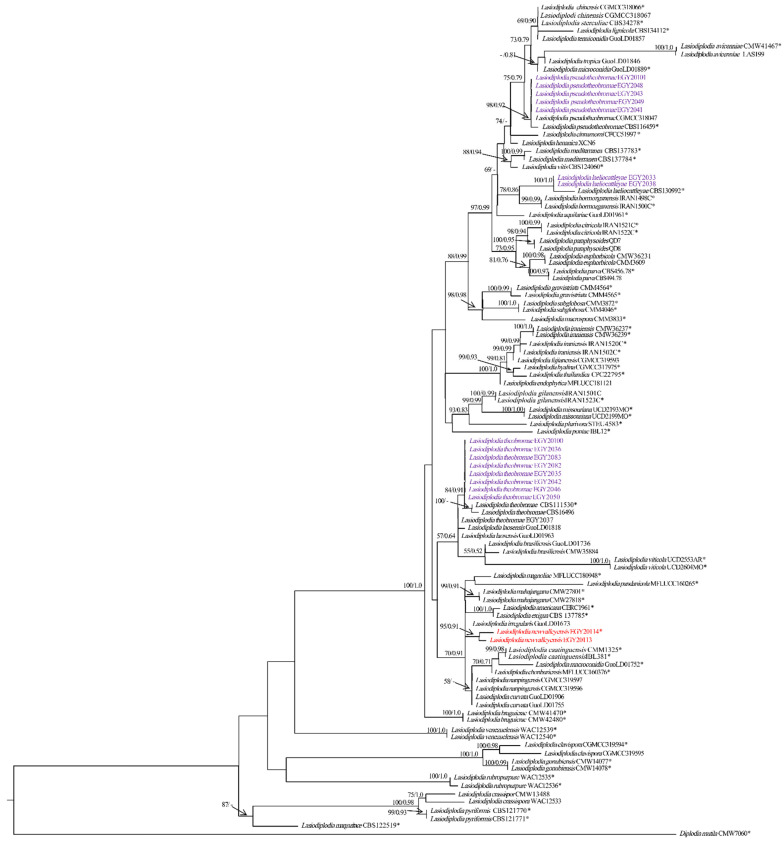
Phylogenetic tree based of maximum likelihood analyses (ML) based on the combined ITS, *tef-1α* and *tub2* dataset of *Lasiodiplodia* species. Branches are shown on nodes with bootstrap values (BS %) and Bayesian posterior probabilities (PP). Branches not supported with BS or PP are marked with –, and isolates representing ex-type are marked with *. *Diplodia mutila* CMW 7060 was used as an outgroup taxon to validate the tree. The isolates obtained in this study are blue, and those newly described and ex-type species are in red boldface.

**Figure 7 jof-08-01203-f007:**
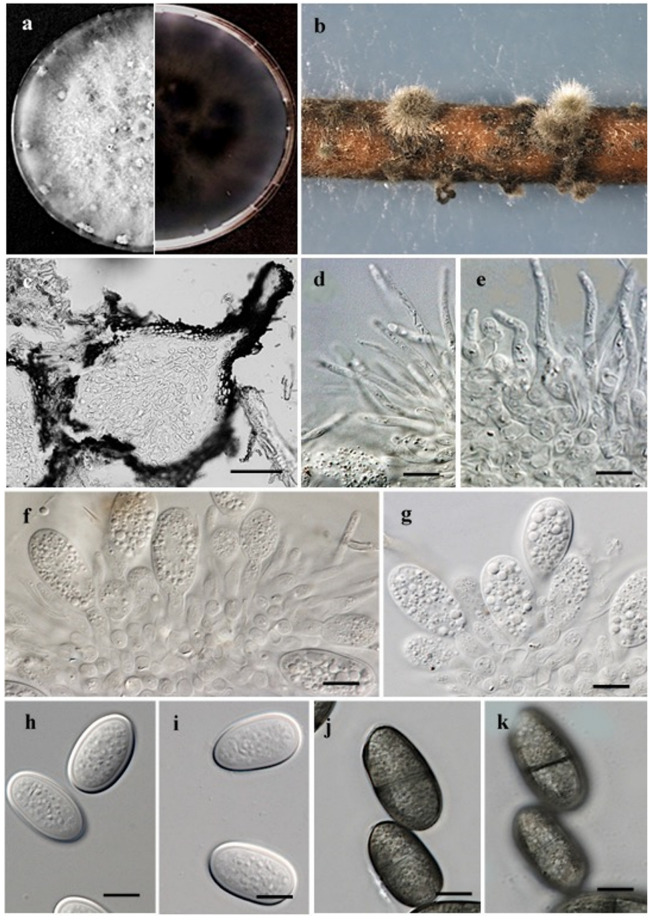
*Lasiodiplodia newvalleyensis* holotype EGY H-240483. (**a**) Colony morphology, front and reverse sides; (**b**) conidiomata formed on pine needles on WA; (**c**) vertical section through pycnidia; (**d**,**e**) hyaline septate paraphyses formed between conidiogenous cells; (**f**,**g**) conidiogenous cells; (**h**,**i**) hyaline immature thick-walled conidia; and (**j**,**k**) dark mature conidia at two different focal planes to show longitudinal striation. Scale bars: (**c**) = 20 µm; (**d**–**k**) = 10 µm.

**Figure 8 jof-08-01203-f008:**
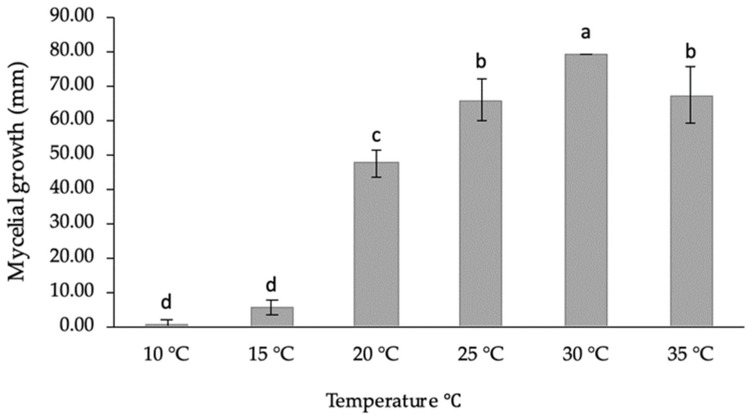
The effect of temperature on the mycelial growth of *L. newvalleyensis* after 3-days on PDA medium. Means followed by the same letter are not significantly different according to LSD test (*p* < 0.05).

**Figure 9 jof-08-01203-f009:**
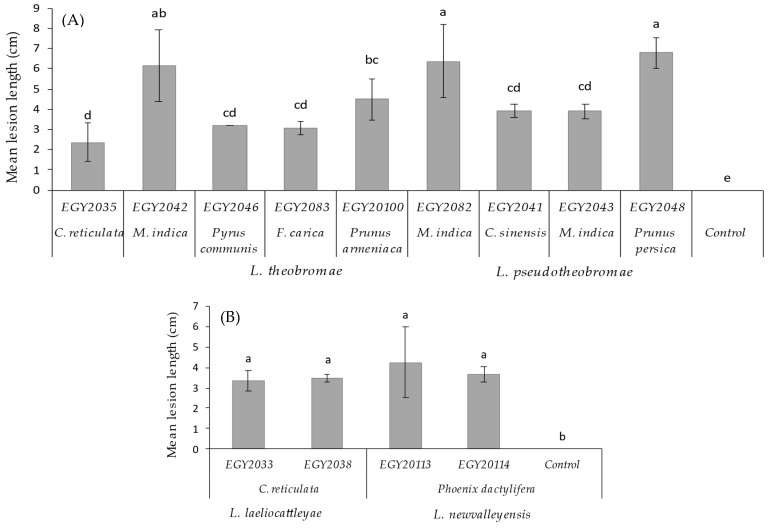
Mean lesion size (mm) (*y*-axis) on stems (**A**) and leaves (**B**) of fruit trees inoculated with 9 isolates (6 of *L. theobromae* and 3 of *L. pseudotheobromae*) and 4 isolates (2 of *L. laeliocattleyae* and 2 of *L. newvalleyensis*) (*x*-axis). Data in these columns are the means of *n* = 9 lesions. Bars above the columns represent standard deviation of the mean. Columns bearing the same letters are not significantly different according to the LSD test (*p* < 0.05).

**Figure 10 jof-08-01203-f010:**
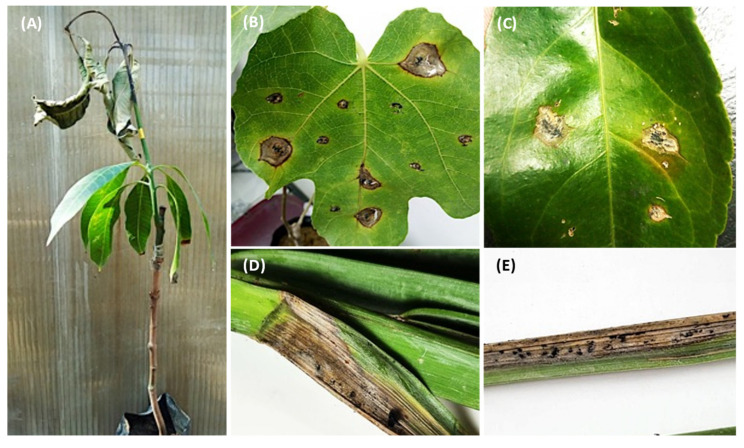
Typical dieback symptoms on mango seedlings after 30 days of inoculation (**A**): necrotic lesions developed around the inoculated tissues of *F. carica* (**B**), *C. reticulata* (**C**) and *Phoenix dactylifera* (**D**)**,** and black pycnidia developed on the necrotic area of *Phoenix dactylifera* (**E**).

**Table 1 jof-08-01203-t001:** *Lasiodiplodia* sequences and their accession numbers used in the phylogenetic analyses.

Species	Strain	Host	Country	GenBank Accession Numbers		
				ITS	*tef1-α*	*tub2*
*L. aquilariae*	GuoLD01961 *	*Aquilaria crassna*	Laos	KY783442	KY848600	-
*L. avicenniae*	CMW 41467 *	*Avocennia marina*	South Africa	KP860835	KP860680	KP860758
*L. avicenniae*	LAS 199	*Avocennia marina*	South Africa	KU587957	KU587947	KU587868
*L. americana*	CERC 1961 = CFCC 50065 *	*Pistachia vera*	China	KP217059	KP217067	KP217075
*L. brasiliensis*	GuoLD01736	*Carica papaya*	Brazil	KY783475	KY848612	KY848556
*L. brasiliensis*	CMW35884	*Adansonia* *madagascariensis*	Madagascar	KU887094	KU886972	KU887466
*L. bruguierae*	CMW41470 *	*Bruguiera* *gymnorrhiza*	South Africa	KP860833	KP860678	KP860756
*L. bruguierae*	CMW42480 *	*Bruguiera* *gymnorrhiza*	South Africa	KP860832	KP860677	KP860755
*L. caatinguensis*	CMM1325 *	*Citrus sinensis*	Brazil	KT154760	KT008006	KT154767
*L. caatinguensis*	IBL381 *	*Spondias purpurea*	Brazil	KT154757	KT154751	KT154764
*L. chinensis*	CGMCC3.18066 *	*Hevea brasiliensis*	China	KX499899	KX499937	KX500012
*L. chinensis*	CGMCC3.18067	*Sterculia* *lychnophora*	China	KX499901	KX499939	KX500014
*L. chonburiensis*	MFLUCC 16-0376 *	*Pandanaceae*	Thailand	MH275066	MH412773	MH412742
*L. cinnamomi*	CFCC 51997 *	*Cinnamomum camphora*	China	MG866028	MH236799	MH236797
*L. citricola*	IRAN1521C *	*Citrus* sp.	Iran	GU945353	GU945339	KU887504
*L. citricola*	IRAN1522C *	*Citrus* sp.	Iran	GU945354	GU945340	KU887505
*L. clavispora*	CGMCC 3.19594 *	*Vaccinium* *uliginosum*	China	MK802166	OL773697	MK816339
*L. clavispora*	CGMCC 3.19595	*Vaccinium* *uliginosum*	China	MK802165	OL773696	MK816338
*L. crassispora*	CMW 13488	*Eucalyptus* *urophylla*	Venezuela	DQ103552	DQ103559	KU887507
*L. crassispora*	WAC12533	*Santalum album*	Australia	DQ103550	DQ103557	-
*L. curvata*	GuoLD01755	*Aquilaria crassna*	Laos	KY783443	KY848601	KY848532
*L. curvata*	GuoLD01906	*Aquilaria crassna*	Laos	KY783437	KY84859	KY848529
*L. euphorbicola*	CMW36231 *	*Adansonia digitata*	Botswana	KU887187	KU887063	KU887494
*L. euphorbicola*	CMW 3609 *	*Adansonia digitata*	Zimbabwe	KF234543	KF226689	KF254926
*L. endophytica*	MFLUCC 18-1121	*Magnolia acuminata*	China	MK501838	MK584572	MK550606
*L. exigua*	IBL 104 = CBS 137785 *	*Retama raetam*	Tunisia	KJ638317	KJ638336	KU887509
*L. fujianensis*	CGMCC3.19593	*Vaccinium uliginosum*	China	MK802164	MK887178	MK816337
*L. gilanensis*	IRAN 1501C	Unknown	Iran	GU945352	GU945341	KU887510
*L. gilanensis*	IRAN 1523C *	Unknown	Iran	GU945351	GU945342	KU887511
*L. gonubiensis*	CMW 14077 *	*Syzygium* *cordatum*	South Africa	AY639595	DQ103566	DQ458860
*L. gonubiensis*	CMW 14078 *	*Syzygium* *cordatum*	South Africa	AY639594	DQ103567	EU673126
*L. gravistriata*	CMM 4564 *	*Anacardium humile*	Brazil	KT250949	KT250950	-
*L. gravistriata*	CMM 4565 *	*Anacardium humile*	Brazil	KT250947	KT266812	-
*L. henanica*	XCN6 = CGMCC 3.19176	*Vaccinium uliginosum*	China	MH729351	MH729357	MH729360
*L. hormozganensis*	IRAN 1498C *	*Mangifera indica*	Iran	GU945356	GU945344	KU887514
*L. hormozganensis*	IRAN 1500C *	*Olea* sp.	Iran	GU945355	GU945343	KU887515
*L. hyalina*	CGMCC 3.17975 *	*Acacia confusa*	China	KX499879	KX499917	KX499992
*L. iraniensis*	CMW 36237 *	*Adansonia digitata*	Mozambique	KU887121	KU886998	KU887499
*L. iraniensis*	CMW 36239 *	*Adansonia digitata*	Mozambique	KU887123	KU887000	KU887501
*L. iraniensis*	IRAN 1502C *	*Juglans* sp.	Iran	GU945347	GU945335	KU887517
*L. iraniensis*	IRAN 1520C *	*Salvadora persica*	Iran	GU945348	GU945336	KU887516
*L. irregularis*	GuoLD01673	*Aquilaria crassna*	Laos	KY783472	KY848610	KY848553
*L. laeliocattleyae*	CBS 130992 *	*Mangifera indica*	Egypt	JN814397	JN814424	KU887508
*L. laeliocattleyae*	**EGY2033**	** *Citrus reticulata* **	**Egypt**	**ON392181**	**OP080238**	**OP080255**
*L. laeliocattleyae*	**EGY2038**	** *Citrus reticulata* **	**Egypt**	**ON392185**	**OP080242**	**OP080259**
*L. laosensis*	GuoLD01818	*Aquilaria crassna*	Laos	KY783471	KY848609	KY848552
*L. laosensis*	GuoLD01963	*Aquilaria crassna*	Laos	KY783450	KY848603	KY848536
*L. lignicola*	CBS 134112 *	dead wood	Thailand	JX646797	KU887003	JX646845
*L. macroconidica*	GuoLD01752 *	*Aquilaria crassna*	Laos	KY783438	KY848597	KY848530
*L. macrospora*	CMM3833 *	*Jatropha curcas*	Brazil	KF234557	KF226718	KF254941
*L. magnoliae*	MFLUCC18-0948 *	*Magnolia candolii*	China	MK499387	MK568537	MK521587
*L. mahajangana*	CMW 27801 *	*Terminalia catappa*	Madagascar	FJ900595	FJ900641	FJ900630
*L. mahajangana*	CMW 27818 *	*Terminalia catappa*	Madagascar	FJ900596	FJ900642	FJ900631
*L. margaritacea*	CBS 122519 *	*Adansonia gibbosa*	Australia	EU144050	EU144065	KU887520
*L. mediterranea*	CBS 137783 *	*Quercus ilex*	Italy	KJ638312	KJ638331	KU887521
*L. mediterranea*	CBS 137784 *	*Vitis vinifera*	Italy	KJ638311	KJ638330	KU887522
*L. microcondia*	**GuoLD01889**	*Aquilaria crassna*	Laos	KY783441	KY848614	-
*L. missouriana*	UCD 2193MO *	*Vitis vinifera*	USA	HQ288225	HQ288267	HQ288304
*L. missouriana*	UCD 2199MO *	*Vitis vinifera*	USA	HQ288226	HQ288268	HQ288305
*L. nanpingensis*	CGMCC3.19597	*Vaccinium* *uliginosum*	China	MK802168	OL773699	MK816341
*L. nanpingensis*	CGMCC319596	*Vaccinium* *uliginosum*	China	MK802168	OL773698	MK816340
** *L. newvalleyensis* **	** EGY20113 * **	** * Phoenix dactylifera * **	** Egypt **	**ON392175**	**OP080253**	**OP080271**
** *L. newvalleyensis* **	** EGY20114 * **	** * Phoenix dactylifera * **	** Egypt **	**ON392180**	**OP080254**	**OP080272**
*L. pandanicola*	MFLUCC 16-0265 *	*Pandanaceae*	Thailand	MH275068	MH412774	-
*L. paraphysoides*	CGMCC 3.19174 = QD7	*Vaccinium uliginosum*	China	MH729349	MH729355	MH729358
*L. paraphysoides*	CGMCC 3.19175 = QD8	*Vaccinium uliginosum*	China	MH729350	MH729356	MH729359
*L. parva*	CBS 456.78 *	Cassava field-soil	USA	EF622083	EF622063	KU887523
*L. parva*	CBS 494.78	Cassava field-soil	USA	EF622084	EF622064	EU673114
*L. plurivora*	STE-U 4583 */CBS 121103	*Vitis vinifera*	South Africa	AY343482	EF445396	KU887525
*L. pontae*	IBL12 = CMM1277 *	*Spondias purpurea*	Brazil	KT151794	KT151791	KT151797
*L. pseudotheobromae*	CBS 116459 *	*Gmelina arborea*	Costa Rica	EF622077	EF622057	EU673111
*L. pseudotheobromae*	CGMCC 3.18047	*Pteridium* *aquilinum*	China	KX499876	KX499914	KX499989
** *L. pseudotheobromae* **	**EGY2041**	** *Citrus sinensis* **	**Egypt**	**ON392168**	**OP080243**	**OP080260**
** *L. pseudotheobromae* **	**EGY2043**	** *Mangifera indica* **	**Egypt**	**ON392170**	**OP080245**	**OP080262**
** *L. pseudotheobromae* **	**EGY2048**	** *Prunus persica* **	**Egypt**	**ON392172**	**OP080247**	**OP080264**
** *L. pseudotheobromae* **	**EGY2049**	** *Mangifera indica* **	**Egypt**	**ON392173**	**OP080248**	**OP080265**
** *L. pseudotheobromae* **	**EGY20101**	** *Mangifera indica* **	**Egypt**	**ON392179**	**OP080252**	**OP080270**
*L. pyriformis*	CBS 121770 *	*Acacia mellifera*	Namibia	EU101307	EU101352	KU887527
*L. pyriformis*	CBS 121771 *	*Acacia mellifera*	Namibia	EU101308	EU101353	KU887528
*L. rubropurpurea*	WAC 12535 *	*Eucalyptus grandis*	Australia	DQ103553	DQ103571	EU673136
*L. rubropurpurea*	WAC 12536 *	*Eucalyptus grandis*	Australia	DQ103554	DQ103572	KU887530
*L. sterculiae*	CBS342.78 *	*Sterculia oblonga*	Germany	KX464140	KX464634	KX464908
*L. subglobosa*	CMM3872 *	*Jatropha curcas*	Brazil	KF234558	KF226721	KF254942
*L. subglobosa*	CMM4046 *	*Jatropha curcas*	Brazil	KF234560	KF226723	KF254944
*L. tenuiconidia*	GuoLD01857	*Aquilaria crassna*	Laos	KY783466	KY848619	KY848586
*L. thailandica*	CPC22795 *	*Albizia chinensis*	China	KJ193637	KJ193681	KY751301
*L. theobromae*	CBS 111530 *	Unknown	Unknown	EF622074	EF622054	KU887531
*L. theobromae*	CBS 164.96	Fruit on coral reef coast	Papua New Guinea	AY640255	AY640258	KU887532
** *L. theobromae* **	**EGY2035**	** *Citrus reticulata* **	**Egypt**	**ON392182**	**OP080239**	**OP080256**
** *L. theobromae* **	**EGY2036**	** *Citrus reticulata* **	**Egypt**	**ON392183**	**OP080240**	**OP080257**
** *L. theobromae* **	**EGY2037**	** *Citrus reticulata* **	**Egypt**	**ON392184**	**OP080241**	**OP080258**
** *L. theobromae* **	**EGY2042**	** *Mangifera indica* **	**Egypt**	**ON392169**	**OP080244**	**OP080261**
** *L. theobromae* **	**EGY2046**	** *Pyrus communis* **	**Egypt**	**ON392171**	**OP080246**	**OP080263**
** *L. theobromae* **	**EGY2050**	** *Pyrus communis* **	**Egypt**	**ON392174**	**OP080249**	**OP080266**
** *L. theobromae* **	**EGY2082**	** *Mangifera indica* **	**Egypt**	**ON392176**	**OP080237**	**OP080267**
** *L. theobromae* **	**EGY2083**	** *Ficus carica* **	**Egypt**	**ON392177**	**OP080250**	**OP080268**
** *L. theobromae* **	**EGY20100**	** *Prunus armeniaca* **	**Egypt**	**ON392178**	**OP080251**	**OP080269**
*L. tropica*	GuoLD01846	*Aquilaria crassna*	Laos	KY783454	KY848616	KY848540
*L. venezuelensis*	WAC 12539 *	*Acacia mangium*	Venezuela	DQ103547	DQ103568	KU887533
*L. venezuelensis*	WAC 12540 *	*Acacia mangium*	Venezuela	DQ103548	DQ103569	KU887534
*L. viticola*	UCD 2553AR *	*Vitis* sp.	USA	HQ288227	HQ288269	HQ288306
*L. viticola*	UCD 2604MO *	*Vitis* sp.	USA	HQ288228	HQ288270	HQ288307
*L. vitis*	CBS 124060 *	*Vitis vinifera*	Italy	KX464148	KX464642	KX464917
*Diplodia mutila*	CMW 7060 *	*Fraxinus excelsior*	Netherlands	AY236955	AY236904	AY236933

* Isolates represent ex-type. The isolates obtained in this study are boldfaced, and those new species are in red boldface.

## Data Availability

All the data related to this study is mentioned in the manuscript.

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
