# Peer review of "Diversity among Lasiodiplodia Species Causing Dieback, Root Rot and Leaf Spot on Fruit Trees in Egypt, and a Description of Lasiodiplodia newvalleyensis sp. nov."

_jof, 2022, doi:10.3390/jof8111203_

Round 1

Reviewer 1 Report

The paper entitled "Genetic Diversity Among Lasiodiplodia species Associated with Dieback, Root Rot and Leaf Spot on Fruit Trees in Egypt” present important information regarding the etiology of some diseases on several fruit trees in Egypt. The introduction is a bit short but clear. The material and methods section is well-written and it contains clear information. The pathogenicity tests are well described, they carefully checked for the presence of normality of their dataset, and the figures are also of great quality. The new species has a well-written taxonomic description. Although the authors have a very detailed comparison of the morphological characters, I have some doubts on the phylogenetic analysis. I recommend adding a table, indicating the nucleotide differences of ITS, tef and tub between the novel species and those that are phylogenetically closely related.

Regarding the discussion, I went through it and I think there are some issues that need to be clarified and better explained. There are also some sentences that seem a bit confusing. I suggest the authors to improve their discussion. The authors gave much attention to the morphological characters that may or not be important in discriminating species in Lasiodiplodia. As this genus contains cryptic species, morphology is not reliable and should always be complemented with phylogenetic analyses. Therefore, in my opinion, the authors must rewrite this part in discussion. Apart from that, the authors do not compare their results with others already published. For example, L. theobromae, L. pseudotheobromae and L. laeliocattleyae have been already reported on mango, blueberries, coconuts, figs, and citrus (10.15741/revbio.06.e595 https://doi.org/10.3767/persoonia.2021.47.03.) and the pathogenicity of these species was also accessed. Therefore, although the present study represents the diversity of Lasiodiplodia in Egypt, it would be interesting if the authors could compare, for instance, the pathogenicity of their isolates with the other studies. Moreover, in the pathogenicity assays section, the authors mentioned that there was a great variation between isolates of L. theobromae and isolates of L. pseudotheobromae in terms of the lesion length. In my opinion, this should be mentioned in the discussion, along with some examples where this was already reported and a possible explanation for this intraspecific variation.

The reference list must carefully be revised. There are some species names that must be in italics and the genus/family names sometimes start with lowercase letters.

I also recommend the authors to revise the manuscript, and to accept my corrections that I added to the pdf version that I attached. I have also some comments I would like you to answer.

Title: I do not agree with the “genetic diversity”. In fact, this study refers only to the diversity of Lasiodiplodia species, and to their genetic diversity. Please rewrite the title.

Line 30: Lasiodiplodia egyptiacae was synonymized as L. laeliocattleyae as you can check at this previous publication: https://doi.org/10.1016/j.funbio.2016.06.004. A more recent publication has no longer used the name L. egyptiacae: https://doi.org/10.3390/life11070657. Please check the manuscript carefully and make the correction at your phylogenetic tree.

Line 57: Verify if you want “multi-locus” or “multilocus”

Line 64-65: Rewrite this sentence. As far as I know, you mention above that L. theobromae has been reported in Egypt. In fact, Ismail et al (2012) carried out a study to evaluate Lasiodiplodia species associated mango in Egypt.

Line 71: Please mention which were the fruit trees you used in your sampling

Line 75: The “et al” you have throughout the manuscript does not need to be in italics.

Line 129: The measurements of the colonies were made after 3 days only? It was when the colonies reached the margins of the plate? Why did you not measure the diameters daily?

Line 134-135: P. persica, P. armeniaca and P. communis? You mean Prunus persica, Pyrus armeniaca and P. communis. It is not clear. You need to be careful. If you have more than one that starts with a P. (for example), you must verify if the genus is the correct. In this case, it is better to mention always the full name.

Line 136-137: It means that you had 12 leaves inoculated with the same isolate? Please clarify the sentence.

Line 139-140: The sentence seems a bit confusing. Please make it clearer.

Line 155: Please highlight in table 1, which of those strains are ex-type strains. Verify also, all the hosts names. If the name was never mentioned before, you must write it all, otherwise it will be confusing to understand.

Line 180-182: Please rewrite these sentences. You need to make a connection between the sentences. This way, it is not clear for the reader.

Line 211: What do you mean with “cv.”?

Line 222: Please verify this species name

Line 228: As already mentioned above, verify the species names that you have on your phylogenetic tree. For instance, the name L. egyptiacae is wrong, and L. jatrophicola is a synonym of D. iranensis. Please verify the phylogeny of Lasiodiplodia and make sure you are using the correct names. Still in Figure 5, highlight the ex-type and/or epitype strains. I also suggest to make a ML analysis of your data.

Line 264: Verify all “Notes” sections and italicize all species names.

Line 296-300: Please mention in what hosts you observed that L theobromae and L. pseudotheobromae were most aggressive, and where the lesions were slightly smaller.

Line 303: You really meat “statistically insignificant”. With a p-value below 0.05?

Line 308: It is better to write the name “Phoenix dactylifera”, as you have been doing this along the manuscript.

Line 315-316: Considering that there is a study on species of Lasiodiplodia occurring in mango in Egypt, you should refer this here.

Line 332: The genus name must be fully written, whenever you start a sentence with the species.

Line 334-337: “but no study has investigated its ecology and distribution”. Where? In Egypt only? Or worldwide. Rewrite this sentence. As the species name is L. laeliocattleyae, there are some studies on the pathogenicity of this species on mango in Egypt (as you mentioned), and Peru (https://doi.org/10.1016/j.funbio.2016.06.004), on mango and coconut in Brazil (https://doi.org/10.1094/PDIS-03-15-0242-RE, https://doi.org/10.1007/s13225-013-0231-z)

Line 353: “the morphological characters reinforce this.” Is it enough to have morphological characteristics to reinforce that you have a potential novel species when you´re dealing with a genus with cryptic species? Please clarify this with the information you have right below.

Line 358-364: This information seems a bit the same. Please clarify it.

Line 360-361: Rewrite the sentence

Line 404: “two new host records” This should be better explained in the manuscript. 

Kind Regards

Reviewer 2 Report

Th author investigated the genetic diversity among Lasiodiplodia species associated  with Dieback, Root Rot and Leaf Spot on Fruit Trees in Egypt using morphological and multi locus phylogenetic approaches. Furthermore, a new species were described. Pathogenicity tests results suggested that all  Lasiodiplodia species were pathogenic to their hosts. The present study is considered the first to characterize and decipher the diversity of Lasiodiplodia species associated with  fruit trees in Egypt, using the ITS, tef1-a and β-tubulin sequence data, along with morphological and pathogenic trials. this work is very interesting. However, several minor suggestions are as below:

1. please added the morphological characters pictures of the similar species,showing the significantly morphological difference with the new species.

2. from the phylogenetic tree, the new species form a subclass but not with high-supported value. It means the bootstrap value(51) or PP value (77) is lower. In generally, the relationship is reliable with the pp value is higher 95%. so, I suggested you conduct the tree using the Bayesian analyses.

3. please implement the one or two pictures of pathogenicity tests results in Figure8

4. please check one by one the spelling of scientific name of species in references!! all of them need to check and correct.

other comments, please see the attachment file.

Round 2

Reviewer 1 Report

The authors have put a great effort to improve the manuscript, and considered my suggestions and corrections. All the manuscript was revised, and the discussion is now clearer.

However, I have some minor suggestion:

Please verify that the tub2 gene is in italics.

Figure 5: It seems that you have 2 figures here. But the caption referes only to one figure. Please revise this. I do not see the ex-type names in the figure highlighted. You did it well in Table 1, but the figures are not corrected. For instance, you can remove the asterisks on bootstrap values. In the case you have */* you can simply delete it. It is implict that the branch has no support. In the case you have 51/*, you can change to 51/-. Thus you can use the asterisk to indicate which strains are the ex-type, following what you have in Table 1. The same is applied to Figure 6.

I also recommend the authors to improve the quality of your trees. They seem a bit blurred. The software Inkscape may help you to save the figures in .eps format for instance.

Figure 7 - There are 2 figures that are overlapped. Please revise this.

Figure 8 - The caption refers to the effect of temperature on the mycelial growth, but you have a second figure about the growth rate. They look like the same. Please revise this.

Figure 9 - You have more than one figure overlapping. Revise this.

In future studies, please consider to compare the nucleotides and the p-distance of all genes, between your potential novel species and those that are phylogenetically closely related. It will give you a more support on the description and veracity of your novel taxa.

Kind Regards

Author Response

Response to Reviewers The authors thank the reviewers for their helpful and detailed comments, respond to each below and have made tracked changes to the script. Comments and Suggestions for Authors Reviewer 1
Reviewer 1
The authors have put a great effort to improve the manuscript, and considered my suggestions and corrections. All the manuscript was revised, and the discussion is now clearer. However, I have some minor suggestion: Please verify that the tub2 gene is in italics. All tub2 changed to italic accordingly. Figure 5: It seems that you have 2 figures here. But the caption referes only to one figure. Please revise this. I do not see the ex-type names in the figure highlighted. You did it well in Table 1, but the figures are not corrected. For instance, you can remove the asterisks on bootstrap values. In the case you have */* you can simply delete it. It is implict that the branch has no support. In the case you have 51/*, you can change to 51/-. Thus you can use the asterisk to indicate which strains are the ex-type, following what you have in Table 1. The same is applied to Figure 6. There are two figures 5 and 6 with two different captions. Figure 5 represents the maximum parsimony analysis while figure 6 represents maximum likelihood analysis. Ex-type isolates had been marked with asterisks. I also recommend the authors to improve the quality of your trees. They seem a bit blurred. The software Inkscape may help you to save the figures in .eps format for instance. The quality of the trees had been improved accordingly. Figure 7 - There are 2 figures that are overlapped. Please revise this.
Checked Figure 8 - The caption refers to the effect of temperature on the mycelial growth, but you have a second figure about the growth rate. They look like the same. Please revise this. Figure 8 had been corrected and the growth rate was changed to mycelial growth because we didn’t measure the rate of the growth over time. Figure 9 - You have more than one figure overlapping. Revise this. Checked In future studies, please consider to compare the nucleotides and the p-distance of all genes, between your potential novel species and those that are phylogenetically closely related. It will give you a more support on the description and veracity of your novel taxa. Thank you for your suggestions and your recommendations will be taken in our consideration in the future studies. Kind Regards

Reviewer 2 Report

The manuscript have significiant improvment, there are only minor spelling error in the Table 1(all species should be italic, e.g. Santalum album  and Vaccinium uliginosum) and references section (the scientific name above genus rank, such as family, order and class etc. It is not necessary to italic (e.g.Botryosphaeriaceae in refernce 1, 41, 43, 53 ,60; Pandanaceae in reference 44 and botryosphaeriales in  58.

Author Response

Response to Reviewers

The authors thank the reviewers for their helpful and detailed comments, respond to each below and have made tracked changes to the script.

Comments and Suggestions for Authors

Reviewer 2

The manuscript have significiant improvment, there are only minor spelling error in the Table 1(all species should be italic, e.g. Santalum album  and Vaccinium uliginosum) and references section (the scientific name above genus rank, such as family, order and class etc. It is not necessary to italic (e.g.Botryosphaeriaceae in refernce 1, 41, 43, 53 ,60; Pandanaceae in reference 44 and botryosphaeriales in 58.

The suggested changes have been done accordingly. Santalum album and Vaccinium uliginosum changed to italic and wherever was applicable. All scientific names above genus rank in reference have been changed to non- italic.

Kind Regards
